# One Health collaboration is more effective than single-sector actions at mitigating SARS-CoV-2 in deer

Jonathan D. Cook [1] ✉, Elias Rosenblatt [2], Graziella V. Direnzo [3], Evan H. Campbell Grant [4], Brittany A. Mosher[2], Fernando Arce[5], Sonja A. Christensen [6], Ria R. Ghai[7] & Michael C. Runge [1]

One Health aims to achieve optimal health outcomes for people, animals, plants, and shared environments. We describe a multisector effort to understand and mitigate SARS-CoV-2 transmission risk to humans via the spread among and between captive and wild white-tailed deer. We first framed a One Health problem with three governance sectors that manage captive deer, wild deer populations, and public health. The problem framing included identifying fundamental objectives, causal chains for transmission, and management actions. We then developed a dynamic model that linked deer herds and simulated SARS-CoV-2. Next, we evaluated management alternatives for their ability to reduce SARS-CoV-2 spread in white-tailed deer. We found that single-sector alternatives reduced transmission, but that the best-performing alternative required collaborative actions among wildlife management, agricultural management, and public health agencies. Here, we show quantitative support that One Health actions outperform single-sector responses, but may depend on coordination to track changes in this evolving system.

The emergence and global spread of severe acute respiratory syndrome coronavirus 2 (SARS-CoV-2) in humans, domestic animals, and wildlife has created a challenging One Health disease mitigation problem. The virus first evolved in a mammalian host, likely Old-World bats of the family *Rhinolophidae*[1], but its transmission to and subsequent spread among the human population has propelled a pandemic of considerable economic, social, and cultural significance. Widespread human transmission has resulted in spillback to numerous mammalian hosts including domestic, farmed, and wild animals (U.S. Department of Agriculture SARS-CoV-2 Dashboard accessed 13-May-2024[2]). Infections of farmed mink (*Neogale vison*) and white-tailed deer (deer, *Odocoileus virginianus*) have caused particular concern in the public health (PH) community because of

documented animal-human transmission and the emergence of novel variants in these species.

Detections of SARS-CoV-2 in deer across four U.S. states began in 2020[3]. Additional surveillance has repeatedly detected multiple strains that are closely related to dominant strains circulating in nearby human populations at the same times[4–6]. A human-collected sample from December 2021 of a highly divergent lineage with mutations associated with non-human hosts was detected in deer and a person with known contact with deer in Ontario, Canada[7]. Legacy strains have also been detected in deer months after they were largely supplanted by new variants within the human population[7]. These findings indicate that human-to-deer transmission has likely occurred repeatedly, and that deer can transmit and sustain SARS-CoV-2 in their populations[6].

[1]U. S. Geological Survey, Eastern Ecological Science Center, Laurel, MD, USA. [2]Rubenstein School of Environment and Natural Resources, University of Vermont, Burlington, VT, USA. [3]U. S. Geological Survey, Massachusetts Cooperative Fish and Wildlife Research Unit, University of Massachusetts, Amherst, MA, USA. [4]U. S. Geological Survey, Eastern Ecological Science Center, Turner's Falls, MA, USA. [5]Department of Environmental Conservation, University of Massachusetts Amherst, Amherst, MA, USA. [6]Department of Fisheries and Wildlife, Michigan State University, East Lansing, MI, USA. [7]U.S. Centers for Disease Control and Prevention, Atlanta, GA, USA. ✉e-mail: jcook@usgs.gov

**Table 1 | Fundamental objectives (i.e., principle agency values) and measurable attributes for risk mitigation decisions across One Health sectors addressing SARS-CoV-2 in AM agencies, PH agencies, and WM agencies**

| Fundamental objectives (measurable attribute) | Sectors |
|---|---|
| Fundamental objective 1. Minimize SARS-CoV-2 infection in humans. (Measurable attribute 1. Median SARS-CoV-2 prevalence in wild and captive deer across the 120-day period) | PH and AM |
| Fundamental objective 2. Maximize the health of deer. (Measurable attribute 2. Median per capita cumulative SARS-CoV-2 infections in wild and captive deer across the 120-day period) | AM and WM |
| Fundamental objective 3. Minimize risk of SARS-CoV-2 evolution. (Measurable attribute 3. The median probability of SARS-CoV-2 persistence in wild and captive deer across the 120-day period) | PH |
| Fundamental objective 4. Maximize deer hunter satisfaction and participation. | AM and WM |
| Fundamental objective 5. Minimize harm to privately owned captive cervid businesses. | AM |
| Fundamental objective 6. Maintain the current management authority of WM, agricultural, and PH agencies. | AM, PH, and WM |
| Fundamental objective 7. Maximize public appreciation, trust, and acceptance of management. | AM, PH, and WM |
| Fundamental objective 8. Provide timely and consistent data-driven measures of local zoonotic disease risks. | AM, PH, and WM |

Measurable attributes (metrics that reflect the achievement of the fundamental objectives) are shown for the first three objectives; the development of the other five metrics was beyond the scope of this paper.

Taken together, it is possible that deer may be a source of novel variants that threaten PH by evading diagnostic detection or affecting therapeutic and vaccine efficacy.

As in other zoonoses, managing SARS-CoV-2 transmission between human and non-human hosts is a challenge that could benefit from a One Health approach[8]. Deer are widespread and abundant in diverse settings across North America, resulting in many possible exposure routes between humans and deer. Furthermore, U.S. management agencies may be essential to participate in solving One Health problems but have governance structures that are complex, with disjunct regulatory authorities that could challenge effective disease management. Understanding linkages across One Health sectors (human, wildlife, and agriculture), and identifying whether coordinated responses are necessary for mitigating disease transmission is a considerable challenge.

An evidence-based approach to inform risk management decisions is essential for effective mitigation in general and for complex governance problems such as One Health, in particular. Therefore, we first coordinated a multisector guidance committee composed of U.S. decision-makers with authority to manage wild deer, captive deer, or PH. This multisector team framed the decisions surrounding the management of SARS-CoV-2 transmission between humans and deer and among deer. The decision framing included an articulation of sector-specific and shared fundamental objectives (i.e., basic values that agencies strive to achieve), identification of causal chains of interactions that may facilitate spillover and spread, and a specification of possible management alternatives (i.e., management strategies). Second, based on a subset of fundamental objectives, we developed and used a deterministic compartmental model to evaluate spillover, spillback, and spread rates in isolated deer herd settings over a 120-day period, and to estimate prevalence, new infections, and persistence in a range of connected human-deer scenarios (the model is described in detail in Rosenblatt et al.[9]). Then, we evaluated a range of single-sector and multisector alternatives (i.e., One Health alternatives that require joint actions by all sectors simultaneously) for their ability to limit the spread of SARS-CoV-2 in white-tailed deer. Collectively, our decision analysis and risk modeling results may assist agencies in setting policy, developing recommendations, and enforcing actions to mitigate the risk of deer-associated SARS-CoV-2 transmission.

## Results

The One Health guidance committee we convened included seven members from wildlife management agencies (WM), five from agricultural management agencies (AM), and five from public health agencies (PH). We used a series of 10, one-to-two hour, meetings to frame the decision context and identify a set of alternatives for evaluation. During the initial meetings, the guidance committee identified eight fundamental objectives to consider in this decision context. In general, the representatives from PH expressed interest in minimizing the morbidity and mortality of coronavirus disease 2019 (COVID-19) in humans by limiting infection (Fundamental objective 1; Table 1). Wildlife and AM agencies expressed interest in maintaining the individual and population health of deer in support of the North American model of wildlife conservation[10] and the Animal Welfare Act (7 U.S.C. 2131 et seq.; Fundamental objective 2; Table 1). Relatedly, all agencies were interested in limiting SARS-CoV-2 transmission within deer herds and other hosts because of the potential for viral strain mutation, recombination with other coronaviruses that circulate endemically in wildlife, and the emergence of novel variants (Fundamental objective 3; Table 1). Novel variants can change the host range, transmissibility, and other viral properties, potentially resulting in new waves of infection within human populations. Thus, efforts to limit disease transmission among deer, humans, and other wildlife species are important. In addition to the disease-related objectives, there were several other objectives expressed among the management sectors related to occupational health, hunter satisfaction, business activities, and management authorities and public trust (Fundamental objectives 4–8; Table 1).

The guidance committee identified two distinct transmission modes for human-to-deer and deer-to-deer transmission: direct transmission (aerosol inhalation or fluids deposited during social contact), and indirect transmission (contact with contaminated fomites or wastewater) (Fig. 1, central component). Within those two transmission modes were seven sources of introduction and spread in wild and captive deer (Fig. 1, green rounded rectangles). For this evaluation, we focused on direct modes of human-to-deer and deer-to-deer transmission based on the highly social behaviors of deer that include nose-to-nose contact and experimental studies that indicate viral replication in the upper respiratory tract and shedding via nasal secretions[11] (Fig. 1, gray dotted line in central rectangle). There is currently no evidence to support indirect transmission modes.

The alternatives considered for the AM sector included requirements to: (1) enhance air ventilation rate in enclosed agricultural/zoo/farm settings for captive deer; (2) double fence captive deer facilities; and (3) vaccinate captive deer (including booster dosing). PH representatives considered two areas that could be the focus of education and training efforts: (1) encourage the public to limit interactions with wild deer in suburban settings (not related to hunting activities) like public parks, and private lands where deer may congregate (e.g., golf courses and residences including yards); and (2) encourage the proper use of enhanced personal protective equipment during interactions with captive and wild deer. For WM, the alternatives included: (1) a pause on permitted research involving close contact with wild deer; (2)

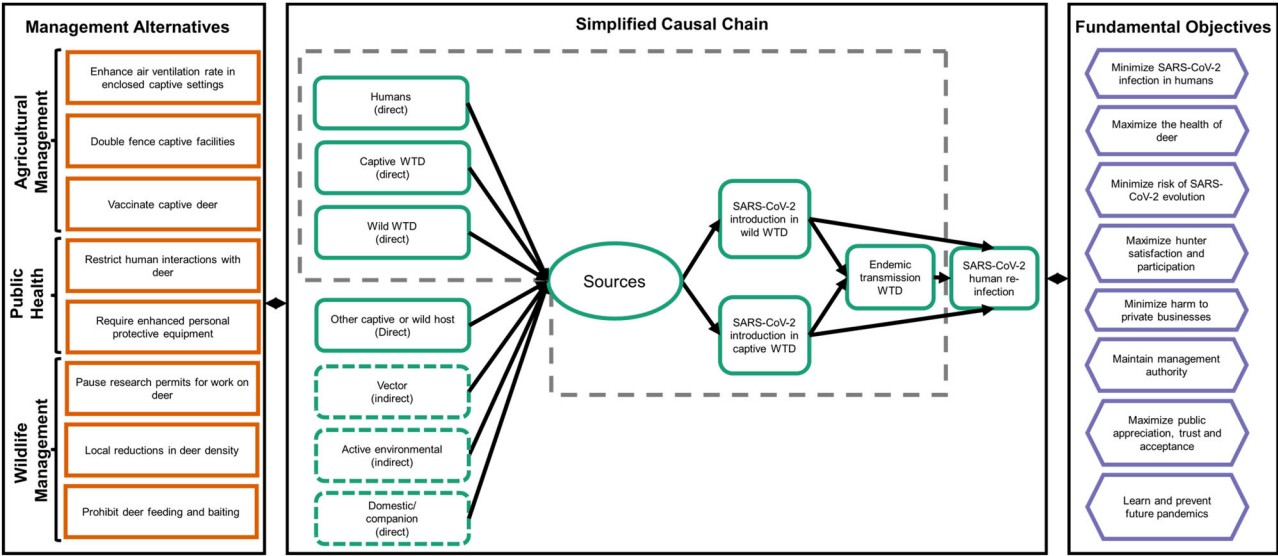

**Fig. 1 | Influence diagram for SARS-CoV-2 introduction and spread in wild and captive deer.** The green ovals show sources of SARS-CoV-2 introduction in wild and captive deer; the orange rectangles show the set of alternatives (i.e., strategies) identified across sectors, and the purple hexagons show the collective set of fundamental objectives (i.e., principle agency values) that each of the three One Health sectors may consider when making risk management decisions. The gray dotted line is the area of the causal chain that we focused on for this risk assessment.

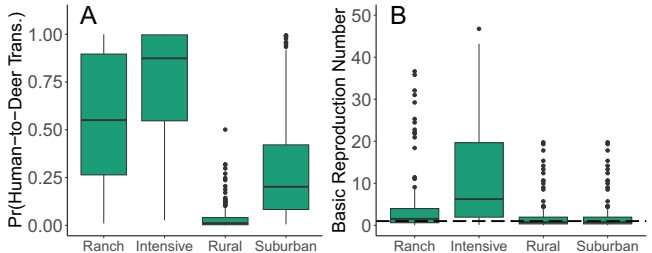

**Fig. 2 | Probability of human-to-deer transmission, and basic reproduction number for deer-to-deer transmission in four captive and wild deer settings.** Boxplots of the probability of human-to-deer transmission (**A**) and basic reproduction number (**B**) in ranch and intensive captive deer herds and rural and suburban wild deer herds. The probability of human-to-deer transmission and the basic reproduction number ($R_0$) were typically higher in captive herd settings. We ran 200 replicate simulations to sample over parametric uncertainty with no initial infections in deer and a continuous hazard of human-to-deer transmission. Pr(human-to-deer transmission) is the probability of at least one human-to-deer SARS-CoV-2 transmission event in 120 days in a completely susceptible population of 1000 deer and across the 200 replicate simulations. The basic reproduction number (**B**) is the average number of secondary cases from an average primary case in an entirely susceptible population. The center line represents the median value, box limits are upper and lower quartiles, whiskers are 1.5 times the interquartile range, and points are outliers.

reductions in wild deer density; and (3) prohibiting wild deer feeding and baiting.

To explore variation in settings where humans and deer may interact, we used a compartmental model to evaluate wild deer in rural and suburban settings and captive deer in low- and high-density facilities (i.e., a total of four settings: rural, suburban, ranch, and intensive). We found that the average probability of at least one human-to-deer transmission event in a population of 1000 deer and across the 120-day window was lower for wild deer than captive deer in each respective setting. The median probability of transmission was 0.01 (95% prediction interval [PI]: $2.2 \times 10^{-4}$–0.25) and 0.20 (95% PI: 0.02–0.96) in rural and suburban wild settings, respectively (Fig. 2). In captive settings, the probability of human-to-deer transmission was

0.55 (95% PI: 0.03–0.99) for ranch deer and 0.87 (95% PI: 0.08–1.0) for intensive captive deer (Fig. 2). For deer-to-deer transmission within wild and captive herd settings, we found that captive herd settings were more likely to have sustained spread over the 120-day period. The median estimate of $R_0$ in intensive captive facilities was 6.24 (95% PI: 0.23–144.0) and in captive ranch facilities it was 1.53 (95% PI: 0.08–25.40, Fig. 2). We found that the median $R_0$ was less than 1.0 in both rural and suburban wild deer settings (median: 0.74 (95% PI: 0.05–13.3)); however, estimates of $R_0$ were highly skewed with many parameter combinations resulting in values greater than 1.0 (Fig. 2).

We then combined settings into scenarios that provided opportunities for captive deer, wild deer, and humans to interact. For the scenarios, we found that baseline estimates of prevalence, per capita cumulative infections (cumulative proportion of the population infected over 120 days), and probability of persistence (the number of model runs where prevalence was greater than 0.001 at the deterministic steady state of the compartmental disease model, divided by the total number of simulations) varied between connected captive and wild deer herds according to combinations of setting types. In general, intensive captive herds interacting with suburban deer and humans had the highest overall prevalence (median: 7%; 95% PI: 0.00–17.90%), whereas rural wild deer interacting with captive ranch deer and humans had the lowest prevalence (median: 3%, 95% PI: 0.00–11.10%) (Fig. 3 and Table 2). For per capita cumulative infections, intensive captive herds again had the highest estimate of 1.46 (95% PI: 0.0001–3.72)) and rural, wild deer had the lowest estimate (0.68; 95% PI: 0.00–2.31; Fig. 3 and Table 2). For the probability of persistence, intensive captive facilities were most likely to sustain the spread of SARS-CoV-2 over the 120-day period (97% of simulations) whereas wild, rural deer were the least likely (64% of simulations; Table 2).

We found consistent patterns in the efficacy of implementing single-sector alternatives at reducing disease burden in deer. For the AM sector, vaccination and boosting captive deer were the most effective alternatives at reducing per capita cumulative infections and the probability of persistence in captive settings (Objectives 2 and 3; Table 2). This intervention also had effects on infections and persistence in sympatric wild deer herds. For PH sector alternatives, we found that increased PPE use during interactions with both captive and

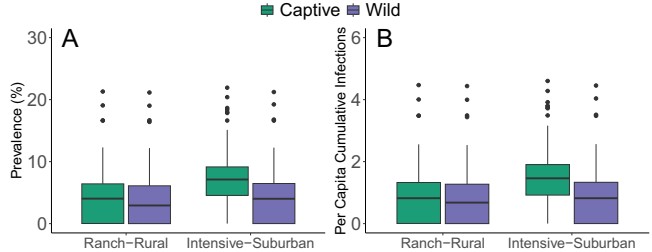

**Fig. 3 | Deer-to-deer SARS-CoV-2 prevalence and per capita cumulative infections in the two scenarios.** Boxplots of prevalence (**A**) and per capita cumulative infections (**B**) in two scenarios of interacting wild and captive deer herds. Prevalence in deer was used as a measure of human health risk, which related to Fundamental objective 1, and per capita cumulative infections were used as a measure of deer health risk, which related to Fundamental objective 2. The center line represents the median value from 200 replicate simulations, box limits are upper and lower quartiles, whiskers are 1.5 times the interquartile range, and points are outliers.

wild deer was the most effective alternative, but did not have the same magnitude of reduction when compared to vaccination and boosting (Table 2). For WM sector actions, we found that both eliminating baiting or feeding and reducing deer densities were more effective at meeting all measurable attributes for wild deer; however, we found little evidence that those same actions would lead to reductions in disease in captive deer (Table 2).

We then considered the effect of coordinated actions by multiple sectors, representing a One Health approach. Thus, we evaluated the outcome when all sectors implemented their own best-performing intervention simultaneously. We found that the combination of vaccination and boosting, the use of PPE in captive and wild settings, and the elimination of baiting for wild deer dramatically reduced the median prevalence, per capita cumulative infections, and probability of persistence in all sectors. This effect was larger than the effect sizes that could be achieved by any one sector acting alone (Table 2); this was particularly true for the suburban and intensive captive deer scenario.

## Discussion

The One Health alternative that included actions by all three sectors was the only alternative that eliminated SARS-CoV-2 transmission in captive deer and substantially reduced transmission in wild deer. In general, this finding supports the critical need for a One Health approach toward improving human, animal, and ecosystem health by jointly considering outcomes in all sectors together, and providing multisector health outcomes that are beyond what single sectors may achieve working on their own. We found that decision analysis was useful in navigating this complex risk and governance problem. By formalizing the problem and providing structure to the decision, this approach helped agencies share information to evaluate SARS-CoV-2 transmission in wild and captive deer and assess the efficacy of alternatives to mitigate it in a way that directly aligns with how agencies view this management problem. This study also highlights the benefits that decision analysis may provide to agencies in navigating simultaneous decision and governance impediments, which we expect can be useful in other disease problems affecting multiple sectors.

While our One Health alternative was drastically more effective than any of the single-sector alternatives, it may be more challenging to implement, as it requires coordination among sectors that have disjunct authorities to manage different parts of the system. Agricultural agencies often have the authority to implement the most effective alternative: vaccination and boosting. This alternative substantially reduced risk within captive deer and provided benefits to

wild deer that may interact along fence lines. While there is currently no vaccine specific to deer against SARS-CoV-2 infection, there are several broad-spectrum vaccines that have proven effective for species that have been inoculated under experimental authority (e.g., Zoetis, Parsippany, NJ, USA). Within the WM sector, reductions in wild deer density and eliminating baiting and feeding reduced disease prevalence and per capita cumulative infections in wild deer. These findings are consistent with other studies that have evaluated the role of density and feeding in intraspecific disease transmission rates[12]. However, a lack of compliance with regulations intended to cease baiting and feeding can limit the efficacy of such measures[13]. Finally, we found that PH guidelines and education may be effective if they can increase the proper use of PPE, particularly if PPE is routinely and consistently used during close human-deer encounters.

Our findings also provide insight into the risks of human-deer spillover and deer-deer transmission in both captive and wild settings under a range of plausible values. While we evaluated conditions that are typical of both deer and human densities in the agro-forested midwestern U.S. (10 deer per km² based on Habib et al.[14] and Walters et al.[15]; 10–100 humans per km² based on U.S. Census Bureau 2020[16]), our models can be altered to consider specific combinations of initial conditions and settings to customize results based on measurements from specific locations. Based on our settings, we found that human-to-deer spillover was highest in intensive captive settings because of elevated proximity rates estimated by an expert panel[9]. For wild deer, the same panel estimated that suburban settings had 2.7 times higher proximity rates when compared to rural and ranch (captive) deer[9]. Despite these differences, Rosenblatt et al.[9] found that simulated disease outcomes had low sensitivity across the range of human-to-deer spillover rates, suggesting that infrequent human-to-deer transmission events are sufficient to initiate sustained outbreaks in deer over a 120-day period. Deer-to-deer transmission, however, was sensitive to parameter values. The estimated range of $R_0$ (0.74 in wild deer and 6.24 in intensive captive deer) was broad and there are many simulations in all settings where $R_0$ exceeded 1.0 (this study and Rosenblatt et al.[9]). These findings suggest that while sustained transmission is thought to occur in many wild and captive deer settings[17], there are likely conditions where deer densities and associated contact rates are insufficient to sustain SARS-CoV-2 outbreaks over a 120-day period.

We begin to address simultaneous decision and governance impediments that are common in One Health settings by clearly articulating some of the motivations, mandates, and decisions that each sector considers in making disease management decisions. However, there remain substantial challenges to fully realizing the benefits of coordinated, One Health actions. First, to fully understand the tradeoffs in different alternatives and identify practical solutions for SARS-CoV-2 spread in the coupled deer-human system, future work would need to evaluate the performance of alternatives across all eight objectives and potential constraints that are unique to each decision setting. For example, even though reducing wild deer density was effective, this alternative may not be practical in all settings, especially in areas with high hunting participation. Second, multiple agencies may need to continue to collaborate and invest resources (e.g., staff time) across multiple spatial scales to realize the benefits of a coordinated strategy. This may be particularly challenging between agencies that have different statutory responsibilities, agency missions, funding mechanisms, and processes for making decisions. Finally, there may be misalignment in the real or perceived benefits that each agency receives from engaging in this One Health problem, such that the costs of intervening might be thought to outweigh the benefits to a particular agency.

Beyond complex governance, there remain large gaps in our understanding of the ecology and epidemiology of SARS-CoV-2. Further, our model did not include details that may be relevant to

**Table 2 | Consequence table showing the effectiveness of sector-specific and One Health alternatives in achieving the fundamental objectives for the two scenarios**

| | AM | | | | | PH | | | | WM | | | One Health |
|---|---|---|---|---|---|---|---|---|---|---|---|---|---|
| | Base | Vent | Fence | Vax | Vax + boost | Interact | PPE capt. | PPE wild | PPE both | No permit | Density | No bait | Vax, PPE, and bait |
| **A. Wild rural and captive ranch scenario** | | | | | | | | | | | | | |
| MA. 1 (cap) | 0.04 | 0.04 | 0.04 | 0.00 | 0.00 | | 0.04 | 0.04 | 0.04 | 0.04 | 0.04 | 0.04 | 0.00 |
| MA. 2 (cap) | 0.82 | 0.82 | 0.82 | 0.00 | 0.00 | | 0.80 | 0.82 | 0.80 | 0.82 | 0.82 | 0.82 | 0.00 |
| MA. 3 (cap) | 0.70 | 0.70 | 0.70 | 0.70 | 0.00 | | 0.67 | 0.70 | 0.67 | 0.70 | 0.70 | 0.70 | 0.00 |
| MA. 1 (wild) | 0.03 | 0.03 | 0.02 | 0.02 | 0.02 | | 0.03 | 0.03 | 0.02 | 0.03 | 0.00 | 0.01 | 0.00 |
| MA. 2 (wild) | 0.68 | 0.68 | 0.57 | 0.57 | 0.57 | | 0.61 | 0.64 | 0.49 | 0.68 | 0.11 | 0.01 | 0.00 |
| MA. 3 (wild) | 0.64 | 0.64 | 0.64 | 0.64 | 0.64 | | 0.64 | 0.62 | 0.61 | 0.64 | 0.6 | 0.45 | 0.43 |
| **B. Wild suburban and captive intensive scenario** | | | | | | | | | | | | | |
| MA. 1 (cap) | 0.07 | 0.05 | 0.07 | 0.03 | 0.00 | 0.07 | 0.07 | 0.07 | 0.07 | 0.07 | 0.07 | 0.07 | 0.00 |
| MA. 2 (cap) | 1.46 | 1.05 | 1.46 | 0.55 | 0.00 | 1.46 | 1.46 | 1.46 | 1.46 | 1.46 | 1.46 | 1.46 | 0.00 |
| MA. 3 (cap) | 0.97 | 0.81 | 0.97 | 0.97 | 0.05 | 0.97 | 0.89 | 0.97 | 0.89 | 0.97 | 0.97 | 0.97 | 0.04 |
| MA. 1 (wild) | 0.04 | 0.04 | 0.04 | 0.04 | 0.04 | 0.04 | 0.04 | 0.04 | 0.04 | 0.04 | 0.03 | 0.01 | 0.00 |
| MA. 2 (wild) | 0.82 | 0.81 | 0.81 | 0.81 | 0.81 | 0.81 | 0.82 | 0.82 | 0.81 | 0.82 | 0.58 | 0.01 | 0.00 |
| MA. 3 (wild) | 0.68 | 0.67 | 0.67 | 0.68 | 0.67 | 0.67 | 0.68 | 0.67 | 0.67 | 0.67 | 0.62 | 0.45 | 0.45 |

Empty cells indicate those where we did not evaluate the alternative with respect to the measurable attribute.

MA measurable attribute, Base baseline, Cap captive, Density reduce deer density, Fence double fence captive deer facilities, MA1 median prevalence of SARS-CoV-2, MA2 median per capita cumulative SARS-CoV-2 infections, MA3 median probability of persistence of SARS-CoV-2, No Bait prohibit feeding and baiting, No Permit pause of permitted research involving close contact with wild deer, Interact reduced human-wild deer interactions, Vent enhance air ventilation rate, Vax vaccinate captive deer.

understanding the best alternative in certain settings, such as the spatial structure of unique landscapes, or sex-specific transmission rates of SARS-CoV-2 in deer. There is also sensitivity to model-based estimates, such as frequency and duration of proximity rates, concentration of SARS-CoV-2 in deer sputum, and deer dose-response relationships (Rosenblatt et al.[9]). Additionally, we lack a full understanding of SARS-CoV-2 host range. The number of mammalian host species that are detected with SARS-CoV-2 continues to increase, and with continued evolution, there is potential for host range shifts or expansions, which may lead to an even greater risk of transmission in the coupled human-wildlife-captive system. Finally, we lack an understanding of how host-pathogen relationships affect intra- and interspecific transmission, spread, and persistence. For example, we assume here that initial human exposure drives SARS-CoV-2 introduction into deer populations, although other routes of exposure, such as from livestock or companion animal sources, are possible but unidentified. Although our knowledge of human-to-human spread has improved, many questions remain around spillover, spillback, and long-term pathogen circulation. These uncertainties and model simplifications may affect risk mitigation decisions and which governing authorities may be best positioned to act. As a result, any multisector attempt to intervene in a rapidly evolving One Health disease problem, such as SARS-CoV-2, may require frequent re-assessment of the ability of interventions to achieve the desired effect that decision-makers are expecting.

One Health action has proven to be successful at mitigating disease in people, animals, and environments for zoonotic diseases like rabies and Rocky Mountain spotted fever[8,18]. However, an ongoing challenge is managing the zoonotic spread of SARS-CoV-2 in humans and wildlife because of the complex ways that humans and deer interact in multiple settings and jurisdictions as well as the widespread infections observed in deer and humans over time[19]. Example applications of decision analysis to One Health issues may be limited currently, yet we believe that there is great potential to use existing methods to collaboratively frame multisectoral disease problems, improve clarity of decision-making within sectors, identify opportunities for resource sharing across sectors, and allow for identifying creative alternatives that enhance One Health outcomes. Here, we applied decision analysis to an important One Health problem. We deconstructed the decision problems of multiple sectors into steps that allowed for a partial evaluation of alternatives unique to each sector. Our results can be used in evidence-based decision-making and support coordinated, One Health efforts to respond to zoonotic diseases, including SARS-CoV-2.

## Methods

### Overview

We sought to frame mitigation decisions surrounding SARS-CoV-2 transmission between humans and deer and among deer using a decision analytical approach[20,21]. We convened a guidance committee composed of three sectors involved with deer management and PH (Supplementary Table 1). We then used a series of facilitated meetings to frame the SARS-CoV-2 decision problem, including specifying the objectives, describing the disease system, and identifying a set of alternatives for evaluation. We provide meeting agendas for our seven facilitated meetings in the Supplementary Methods. This research conformed with guidelines and recommendations, included a diversity of backgrounds and expertise on our committee and expert panels, and required no handling of vertebrate animals or human subjects.

### Components of decision framing

The first step was to articulate the scope of the problem, including the spatial extent and specific authorities of agency representatives. The guidance committee defined the spatial scope of the evaluation as

regions in the U.S. with high wild deer density, where captive deer occur, and with human populations in proximity to both wild and captive deer (e.g., agro-forested midwestern states). In terms of authorities, state WM agencies in the U.S. have primary authority to manage wild deer within their boundaries; however, other federal agencies have shared authority on certain federal lands. In most U.S. states, captive deer are managed by state agricultural agencies, including farmed deer and deer on exhibition. In some locations, these authorities are shared with state WM agencies. Animals kept in captivity and those that are involved in the trade are regulated in a manner that is consistent with their intended use. PH agencies make decisions to promote health at all governance levels, including federal, state, tribal, local, and territorial health departments. In the context of this study, PH agencies use their authorities to provide PH guidance and recommendations, compile data, and communicate zoonotic disease detections so that at-risk groups can practice individual risk mitigation. Federal PH agencies collaborate with state, tribal, local, and territorial jurisdictions, which investigate illness and disease, set reporting requirements, and notify the public about disease threats in their area.

We then worked on framing the specific goals (i.e., fundamental objectives), the causal chains of interactions that may affect SARS-CoV-2 transmission, and interventions to interrupt transmission. We summarized this information using an influence diagram to visualize and map connections among fundamental objectives and alternatives following the causal chains[22].

Finally, we used the influence diagram to generate a full set of alternatives (we provide a full list of alternatives in the Supplementary Materials), and then pared back the list to include a subset of alternatives of greatest interest and perceived utility to committee members, representing management agencies.

## Evaluation of the one health decision problem

We then developed a deterministic compartmental disease model that could test the efficacy of these alternatives in a simulated system where humans, wild, and captive deer could interact (Rosenblatt et al.[9] for details ref. [9]). The model was designed to evaluate the disease-related objectives with the expectation that, if effective One Health alternatives were identified, future analyses could estimate the effect on the remaining objectives given the context where specific actions may be taken. The model allows captive and wild deer to transition among susceptible ($s$), infectious ($i$), and recovered ($r$) compartments before eventually returning to the susceptible compartment. The model estimates the proportions of individuals in each disease compartment, does not include demography, and assumes a closed population (i.e., no births, deaths, immigration, or emigration). Given the assumption of a closed population during a time of harvest, we must also assume that harvest occurs randomly and that individuals in susceptible, infectious, and recovered compartments are harvested with equal likelihood. Transmission occurs in deer from interacting with infectious humans or deer in captive and wild herds. Deer are assumed to be evenly distributed with all individuals subjected to the same contact rates. We parameterized the model to simulate the midwestern U.S. during the fall season (September-December; 120-day window) when humans may contact deer during hunting and other activities. We acquired parameter estimates using empirical data and expert elicitation[9] (for a full description of the expert elicitation process see ref. [9]). The decision to simulate across 120 days was guided by a panel of virology experts and designed to reduce the potential for evolutionary or ecological changes that may affect our results.

To explore variation in settings where humans and deer may interact, we evaluated wild deer in rural and suburban settings and captive deer in low- and high-density facilities (i.e., a total of four settings: rural, suburban, ranch, and intensive). For model simulation purposes, we defined a rural setting as a free-ranging deer population at a density of 10 deer/km$^2$,[13] in a 26% forested landscape with a human density of 3.1 humans/km$^2$. For a suburban deer setting, we maintained the same deer density and forested cover, but we increased human density to 100 humans/km$^2$ and increased the human-deer contact rate. For the captive deer settings, we defined a low-density condition equal to the density of rural wild deer but assumed an elevated human-deer contact rate (e.g., from supplemental feeding and other herd maintenance activities). We refer to this low-density captive deer setting as the 'ranch'. Lastly, we defined a high-density setting ('intensive') as a herd in an enclosed environment and assumed less air exchange and elevated deer-to-deer and human-to-deer contact rates relative to the ranch setting[9].

To understand model performance and baseline conditions prior to evaluating alternatives, we first estimated disease introduction (from humans) and spread in the four settings. We ran 200 replicate simulations to sample over parametric uncertainty with no initial infections in deer and a continuous hazard of human-to-deer transmission. We summarized results as the probability of at least one human-to-deer SARS-CoV-2 transmission event in 120 days in a completely susceptible population of 1000 deer and the basic reproduction number, $R_0$, for deer-to-deer transmission (i.e., average number of secondary cases from an average primary case in an entirely susceptible population[23]).

We then combined these settings into scenarios that allowed captive deer, wild deer, and humans to interact. We explored two scenarios: (1) interacting with rural wild deer, farmed ranch deer, and rural human populations (rural × ranch), and (2) interacting with suburban wild deer, captive intensive deer, and suburban human populations (suburban × intensive). Based on human-deer and deer-deer contact rates, we designed these combinations to explore a lower-risk scenario (scenario 1) and a higher-risk scenario (scenario 2).

To evaluate the performance of the alternatives on disease-related objectives (objectives 1–3) in the two scenarios, we evaluated three different measurable attributes for each wild and captive deer, producing a total of six measurable attributes (Table 1). The measurable attribute for objective one was the median prevalence of SARS-CoV-2 in wild and captive deer across all simulations. Each simulation was summarized by the average daily proportion of the population in the infectious compartment. We assumed that a higher prevalence in deer would elevate the potential for any contact between humans and deer to have a higher risk of deer-to-human transmission. For objective two, we used per capita cumulative infections of SARS-CoV-2 in wild and captive deer. This metric was calculated as the median of the summed total proportion of daily infections during each 120-day simulation. Per capita cumulative infections can be greater than 1 which represents an outcome where more individuals were infected multiple times than individuals that were not infected within a single simulation[9]. We assumed that per capita cumulative infections were a good measure for deer health outcomes because this metric quantified the degree of spread within a population during 120 days, including the potential for immunity to wane and for re-infection to occur. Finally, for objective three, we used the median probability of persistence of SARS-CoV-2 in wild and captive deer because we assumed that persistence of disease would elevate the risk of evolution because extended periods of transmission would result in more opportunities for mutation and recombination within and among infected deer. We defined persistence as any replicate with a proportion in the infectious compartment at equilibrium (i.e., the deterministic steady state of the compartmental disease model) that exceeded 0.001. The goal was to identify alternatives that could minimize estimates of each measurable attribute. The complete set of results for each of the two scenarios is reported in a consequence table (Table 2), which summarizes the performance of each alternative relative to the objectives considered. For an in-depth explanation of the model, its parameterization,

including the implementation of alternatives and quantitative description of metrics, see ref. 9 and Rosenblatt et al.[24].

## Reporting summary

Further information on research design is available in the Nature Portfolio Reporting Summary linked to this article.

## Data availability

All data used in this study can be simulated using the R software package whitetailedSIRS. Citation: Rosenblatt, E. et al.[24], https://doi.org/10.5066/P9TZK938. Personal notes and transcripts taken during the guidance committee meetings are not publicly available owing to privacy concerns. For more information, contact Jonathan Cook (jcook@usgs.gov) and allow two weeks for a response.

## Code availability

The code used in this study is available as an R software package: whitetailedSIRS. Rosenblatt, E. et al.[24], https://doi.org/10.5066/P9TZK938.

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

## Acknowledgements

We thank Margaret McEachran (University of Massachusetts), Jennifer Mullinax (University of Maryland), and Michael Tonkovich (Ohio Department of Natural Resources) for their comments and suggestions on previous versions of the manuscript. We thank Casey Barton Behravesh (U.S. Centers for Disease Control and Prevention), Colin Basler (U.S. Centers for Disease Control and Prevention), Samantha Gibbs (U.S. Fish and Wildlife Services), Colin Gillin (Oregon Department of Fish and Wildlife Resources), Allen Gosser (U.S. Department of Agriculture Wildlife Services), Chelsea Gridley-Smith (National Association of County and City Health Officials), Paul Johansen (West Virginia Department of Natural Resources), Darlene Konkle (Wisconsin Department of Agriculture, Trade and Consumer Protection), Roxanne Mullaney (U.S. Department of Agriculture Animal and Plant Health Inspection Services), Darby Murphy (U.S. Fish and Wildlife Services), Susan Rollo (Texas Department of State Health Services), Lisa Shender (National Park Service), Jennifer Siembieda (Food and Agriculture Organization of the United Nations), Jason Sumners (Missouri Department of Conservation), Michael Tonkovich (Ohio Department of Natural Resources), and Nora Wineland (Michigan Department of Agriculture and Rural Development). This work was supported by the Coronavirus Aid, Relief, and Economic Security Act (P.L. 116-136). Any use of trade, firm, or product names is for descriptive purposes only and does not imply endorsement by the U.S. Government. The opinions expressed by authors contributing to this journal do not necessarily reflect the opinions of the Centers for Disease Control and Prevention or the institutions with which the authors are affiliated, but do represent the views of the U.S. Geological Survey. This is publication #06 of the Disease Decision Analysis and Research (DDAR) group of the U.S. Geological Survey.

## Author contributions

J.D.C., E.R., G.V.D., E.H.C.G., B.A.M., F.A., S.A.C., R.R.G., and M.C.R. conceived of the study. J.D.C., E.R., E.H.C.G., and M.C.R. developed the methods and performed the analysis. J.D.C. prepared the original draft manuscript. All co-authors reviewed and edited the final draft for submission.

## Competing interests

The authors declare no competing interests.
