## [Peer Review File · Nature Communications]

One Health collaboration is more effective than single sector actions at mitigating SARS-CoV-2 in deerREVIEWER COMMENTS

Reviewer #1 (Remarks to the Author):

This study provides valuable insights into the importance of multisector cooperation to mitigate emerging and zoonotic infectious disease threats. The study exemplifies a One Health approach by integrating different sectors such as wildlife management, agriculture, and public health. This holistic perspective is crucial for understanding and managing zoonotic diseases that can cross between animals and humans.

However, the current version of the manuscript raises several concerns that must be addressed to make it relevant and maximize the value of this report for a broader readership.

Major comments:

1. As the authors have pointed out, several unanswered questions remain regarding the ecology and epidemiology of SARS-CoV-2, including the role of fomites and environmental sources such as wastewater and peridomestic animals like deer mice in transmitting the virus to deer. Research in these areas and new knowledge, as it becomes available, could conceivably change the necessary mitigation steps and interventions. However, it remains unclear how the multisector implementation of these mitigation strategies will incorporate the new knowledge and adapt their approaches accordingly. Given that the proposed solution involves each agency implementing steps simultaneously without a coordinating body to structure their efforts, the authors do not comment on how the new insights will be integrated and action steps amended, and who will be responsible for this. This represents a major flaw and a missed opportunity.

2. The analysis makes several assumptions that simplify a complex RNA virus disease problem involving multiple host species. For example, the assumption that vaccinating captive deer will effectively prevent infection overlooks the complexity of viral evolution and the ecological dynamics of deer populations. The impact of new SARS-CoV-2 viral variants in humans and the turnover in deer populations, which could compromise the vaccine's efficacy, needs to be considered.

While the study addresses a specific disease problem within the U.S., it lacks a broader discussion on how these findings could inform One Health initiatives for other diseases both domestically and internationally.

3. Including academic partners representing virus and animal health research experts could have provided a more comprehensive analysis and improved the model's accuracy and relevance.

Minor comments:

1. The repeated use of "human public health" in multiple sections (for example, lines 125 and 148) is redundant. Since "public health" inherently involves human health, simply using "public health" is sufficient and clearer.

2. In line 171, the first use of the abbreviation 'COVID-19' should include its full description.

Reviewer #2 (Remarks to the Author):

Thank you for the opportunity to review this manuscript. I found the approach and the results interesting, and generally enjoyed reading the manuscript. The article provides an example of the benefits of a coordinated One Health approach to disease management, demonstrated through modeling spillover of SARS-CoV2 from human-to-deer and amplification within deer populations. I think this is a useful contribution. However, there were some issues that will need to be addressed before publication.

My main concerns are related to the adequacy of the model to meet all the challenges presented in the stakeholder engagement section of the manuscript. Clearly, the model is only addressing a fraction of the "fundamental objectives" identified by each sector, which creates a little bit of a disconnect between the two sections. If the model ignores 5 of 8 fundamental objectives, is it really helpful in exploring the decision space and identifying tradeoffs. The model was, however,

very useful in exploring the added value of multi-sectorial management strategies compared to mono-sectorial (the main contribution of this article) in addressing the risk of spillover and amplification. The manuscript should better articulate, the connection between these two sections, and acknowledge and justify that the model is only addressing a narrower scope. Further, recognizing the scope of the model, it was not well justified how the measurable attributes were actually aligned with fundamental objectives 1-3. I indeed suspect that they may not always be good proxies for the stated objectives (see detailed comments below). This may all be addressed by using appropriate terms, and stating transparently what the model can and cannot do. i.e with a 120 day simulation, I am not entirely sure that you can assess long-term maintenance in deer populations, however, you can assess initial spillover and amplification (rather than long-term persistence).

The detailed comments below include both minor edits and more substantial comments that align with this general statement:

L48: replace "helps" with "aims to"

L50: Does "across OH sectors" refer to effort or spread? Clarify sentence

L50: revise term "risk problem"

L51: do you need "human" in front of public health?

L52: Framing of what?

L58: Here and throughout, I suggest the use of term strategies instead of "alternatives"?

L118: May need to be a little more focused, do you mean health and natural resources, or all agencies?

L124: You have used the term sector before, but jurisdictions here... do you mean the same here or ...

L125: What does "This decision framing" refer to?

L131: Is "One Health collaborative" referring to the same multi-jurisdiction team? I suggest to call it something and to stick to that name to avoid confusion.

L142-145: Move this to discussion

L145: is this another term for the same group?

Suppl table 1: It appears that PH mostly had federal folks involved... what is the effect of that on the committee?

L149: again, I believe the word "strategies" would be more appropriate here?

L165: I suggest to keep the use of the term "framing" consistent in your manuscript. If you describe decision framing as the process of identifying potential strategies, then stick to that and use alternate language to describe other processes. Here you could just replace "framing" with "describing"

L176-179: This would typically be more of a discussion statement

L169-176 and L179-181: This should be in your result section.

Table 1: How is "median SARS-cov2 incidence in deer across 120-day period" an indication of infection in humans? I understand that you are trying to match sector-specific objectives with measurable attributes of the mode, but this does not seem like a good match. Being realistic about what the model can and cannot assess is important for this work.

Similarly, not quite sure that a 120 day simulation supports this objective. As noted further down, This model is really focused on spillover and amplification, but is not suitable to assess maintenance within deer populations. Spillover/amplification are extremely important in the process of establishing new reservoirs, and worthwhile assessing with a model, but the authors should be more realistic about what the model can and cannot measure/estimate.

Fundamental objective 4-8 are absolutely not addressed in the model and paper...How is the process and model supporting a One Health approach if 5 out of 8 fundamental objectives are not taken in consideration?

L206: replace "of committee members" with "to committee members"

L221-223: Ignoring demography suggest you are not interested in the long-term persistence of SARS-CoV2 in deer populations, which seems counter to the ideas discussed above related to deer acting as reservoirs for the virus.

L227: Is it reasonable to assume a closed population during hunting season? Please justify

L244: It was not immediately clear in the objective description that you were going to explore risk uncertainty by simulating across the parameter uncertainty? This may need to be clarified earlier in the objective statement

L247: may be useful to specify that this refers to deer-to-deer R0.

Supplementary Method:

- Missing justification for the time scale of the model... why is it limited to 120 days?
- Inconsistency in description of mode of transmission: the initial text refer to direct and indirect, but there is no reference to indirect transmission in the rest of the document (instead fluid transmission is also referred to as direct)
- reference to equation numbers are not consistent with equation description (equation not numbered)
- there is no information on how the number of infectious humans is set: I understand it was only considered as a source of infection and not explicitly modeled, but how was it set? A fixed proportion, was it allowed to fluctuate? What was the source of data for it?

L258: At this point of the manuscript, it is not clear how the "alternatives" are included in the model

L259-263: As indicated above in Table 1: The relevance of measurable attributes to each objectives need better justification. I am not convinced that your measurable attributes are actual indicators of your outcome of interest (i.e. objectives)

L263: Is this evidence of persistence or amplification? What is the evidence that you actually reach an endemic equilibrium? Non-zero prevalence only means that you have continued transmission since the initial spillover and until end of simulation (i.e. persistence ≤ 120 days), it doesn't necessarily mean that the virus will persist endemically in the population.

L264: How did you assess that simulations had reached equilibrium?

L266: "summarizes"

L266-267: I think you mean "performance of each alternative relative to the objective considered"

L267: are these tradeoffs? You are not showing any negative effects of any of these interventions, so not sure the term tradeoff is appropriate here.

Fig 2: I am wondering if this would not be better presented as a 3 dimensional graph, where Pr(H-to_D) and R0 are the main two dimensions, and your third dimension are the different contexts... this would be show the interplay of risk of spillover and risk of subsequent transmission/amplification, similar to the conceptual diagram in Viana et al 2015... Just a suggestion as it may make the graph harder to read

L297: Clarify your definition here.

Cannot be both a proportion and a number of infections. I think you mean the proportion of the population newly infected during the season. Technically, an incidence proportion (cumulative incidence) should be between 0 and 1. I understand that in this case, you have re-infections, but a more appropriate measurement may be an incidence rate that take into account the appropriate number of individual at risk over time.

L305: You need to provide a clearer definition of Probability of persistence.

Table 2: There is nowhere in the manuscript where you actually explain what the values in the table are. This needs to be better described, how did you derive these values of relative effectiveness? The color shading is also not consistent with the the values, which adds confusion.

L351-353: The promise of OH is about the synergy between sector. In this context, the expectation is that the outcome of joint action is greater than the additive effects of sector-specific actions... I think it would be worth highlighting in your discussion

L376-380: There should be more specific statements regarding the limitations of your model, particularly in regard to simplifying assumptions (e.g. lack of spatial structure, lack of population heterogeneity etc)

L382: There is very little information on how model parameters were elicited in this specific study, which should be better addressed in the material and methods

L385-386: The sensitivity analysis is greatly missing from this article, particularly as a way to assess the effect of parameter choice on your inference regarding different contexts and scenarios.

L394: I am not clear how the manuscript is doing this? Yes, you articulated objectives of each sector, but I am not clear what contradictions were perceived in these objectives.

L399-401: Does this mean this work missed an opportunity to address actual barriers to decision that reach intended outcomes? Are you going to repeat this work? were mechanisms put in place that will facilitate addressing these issues?

L404-412 should be better referenced with current literature.

Reviewer #3 (Remarks to the Author):

Overall, I find the article relevant, well constructed, and clearly written.

The main result is that adopting a One Health strategy of multi-sectoral expertise and prioritisation, and simultaneous actions to reduce the transmission of SARS-CoV-2 among humans and white-tailed deers, is more efficient than non-coordinated sectoral actions.

The paper provides a quantitative estimate of the impact of the different strategies of action (coordinated or not), based on a model built on the multi-sectoral expertise. It also implements a method to produce multi-sectoral expertise, as well as prioritisation of possibles measures. It should emphasise better its methodology as a contribution.

The work is relevant to the field of the governance of infectious diseases that display contaminations between humans and domestic / farm / wild animals.

The fact that coordinated interventions are more efficient than non-coordinated ones is expected. However, the demonstration contributes to raising the standard for demonstrating the efficacy of the One Health approach. Indeed, the claim that One Health strategies are more efficient than non-coordinated sectoral strategies is frequently made at the international and national levels, but rarely estimated precisely at the sub-national level.

The work supports the conclusions and claims.

The text would gain from expressing more clearly, at the start of the paper, how the authors define a One Health strategies, since it first appears in l. 330-331 ("The One Health alternatives (i.e., joint actions taken by all sectors simultaneously) led to the best outcomes" and l. 339-341, "We then considered the effect of coordinated actions by multiple sectors in line with a One Health approach. Thus, we evaluated the outcome when all sectors implemented their own best performing intervention simultaneously").

The authors highlight the benefits of a coordinated One Health strategy, but the text does not discuss the costs and difficulties to design and implement it. This is not the core of the paper, and is difficult to make, but extra thoughts (and data, if applicable) on this would give more breadth to the paper. In the same vein, I suggest to change l. 115-116 "Like other zoonoses, managing SARS-CoV-2 transmission between human and non-human hosts is a challenge that requires a One Health approach" -> "... that could benefit from a One Health approach", since the benefit is what is to be demonstrated rather than claimed at the beginning.

The number of workshops, the selection of the participants, and the organisation of the discussions, constitute a sound and coherent methodology.

Epidemiological modelling is outside my area of expertise, therefore I cannot provide relevant feedback on this section. However, it is not the core of this paper.

The concepts used would gain from being made more explicit at the beginning of the text. One main object of the article is governance, for a pluridisciplinary audience, but the writing uses concepts (such as "framing", "alternatives") with a meaning that is much more frequent in epidemiology than in political science and could be clarified to ease the reading.

The methods are generally well presented. The availability and documentation of the code are excellent.

The methodology used to run the meetings with the stakeholders should be explained in greater detail.

How were the discussions structured? Which methodology was used to get to consensual results? For instance, sentences such as l. 165-166 "We then worked on framing the specific interests (i.e. objectives), the causal chains of interactions that may affect disease transmission, and interventions to interrupt transmission" could gain from explaining how the work mentioned was done.

Reviewer #3 (Remarks on code availability):

The code is presented as an R package, with excellent documentation. I could install and run the code. I did not test everything, but what I tested worked flawlessly.

REVIEWER COMMENTS

Reviewer #1 (Remarks to the Author):

This study provides valuable insights into the importance of multisector cooperation to mitigate emerging and zoonotic infectious disease threats. The study exemplifies a One Health approach by integrating different sectors such as wildlife management, agriculture, and public health. This holistic perspective is crucial for understanding and managing zoonotic diseases that can cross between animals and humans.

However, the current version of the manuscript raises several concerns that must be addressed to make it relevant and maximize the value of this report for a broader readership.

Major comments:

1. As the authors have pointed out, several unanswered questions remain regarding the ecology and epidemiology of SARS-CoV-2, including the role of fomites and environmental sources such as wastewater and peridomestic animals like deer mice in transmitting the virus to deer. Research in these areas and new knowledge, as it becomes available, could conceivably change the necessary mitigation steps and interventions. However, it remains unclear how the multisector implementation of these mitigation strategies will incorporate the new knowledge and adapt their approaches accordingly. Given that the proposed solution involves each agency implementing steps simultaneously without a coordinating body to structure their efforts, the authors do not comment on how the new insights will be integrated and action steps amended, and who will be responsible for this. This represents a major flaw and a missed opportunity.

This is a great point, and one that we overlooked in the drafting of this manuscript. The decision science approach we used is iterative and provides a way to incorporate new information as it becomes available. However, the question of who is responsible for leading this integration and revision is complicated and will certainly vary across disease problems, funding, the nature of the new information, and stakeholder engagement. Our goal in this paper is to present the initial prototype for the SARS-CoV-2 decision problem that we framed. Based on your comments, we have added text to highlight the importance of ongoing coordination, both at the end of the abstract (line 68) and in the discussion (lines 504-506).

(line 68) “Thus, our study highlights the potential for One Health actions to enhance single sector responses to SARS-CoV-2 but may depend on ongoing coordination to track and integrate changes in this rapidly evolving system.”

(lines 504-506) “As a result, any multisector attempt to intervene in a rapidly evolving One Health disease problem, such as SARS-CoV-2, may require frequent re-assessment of the ability of interventions to achieve the desired effect that decision makers are expecting.”

2. The analysis makes several assumptions that simplify a complex RNA virus disease problem involving multiple host species. For example, the assumption that vaccinating captive deer will effectively prevent infection overlooks the complexity of viral evolution and the ecological dynamics of deer populations. The impact of new SARS-CoV-2 viral variants in humans and the turnover in deer populations, which could compromise the vaccine's efficacy, needs to be considered.

While the study addresses a specific disease problem within the U.S., it lacks a broader discussion on how these findings could inform One Health initiatives for other diseases both domestically and internationally.

We agree that the disease problem is complex, dynamic, and likely to change over time in ways that are difficult to predict, especially given the high rates of recombination and evolution observed in coronaviruses. We discussed these concerns with an expert panel of virologists. Based on those conversations, we decided to limit the temporal scope to 120 days to limit unexpected changes in disease dynamics during that period. We have added detail on this point at lines 278-280.

As for broader relevance to other domestic and international concerns – we agree that our methods and findings may provide some useful information. We have tried to include broader implications in the last paragraph of the discussion, including moving a sentence that was previously out of place in the methods.

3. Including academic partners representing virus and animal health research experts could have provided a more comprehensive analysis and improved the model's accuracy and relevance.

We agree that including a diversity of partners helps provide a comprehensive analysis and improves model accuracy. In addition to our guidance committee, modeling team, and co-authors on the paper who hold years of experience and expertise in the areas required for modeling zoonotic diseases, we also convened two expert panels. These panels represented viral and animal health research experts and helped us with estimates of important but unknown parameter values that were necessary for our epidemiological model.

The panels included the following experts: Paul Cross (USGS Disease Ecologist), Anna Fagre (Colorado State Research Scientist), Kate Huyvaert (Washington State University disease ecologist), Jeff Root (USGS Research Wildlife Biologist), Jeff Chandler (USDA microbiologist), Sarah Hamer (Texas A&M veterinary ecologist), Kamen Campbell (Mississippi Dept of Wildlife, Fisheries and Parks assistant deer program leader), Chris Jennelle (Minnesota Department of Natural Resources biometrician), Jonathan Trudeau (Maryland Department of Natural Resources game mammal section leader), Kurt Vandegrift (Penn State University senior research associate), and Noelle Thompson (Western Association for Fish and Wildlife Agencies disease coordinator). The names and affiliations of these experts is included in the Supplementary information of Rosenblatt et al. (2024) in PLoS Computational Biology.

Citation: Rosenblatt E, Cook JD, DiRenzo GV, Grant EHC, Arce F, Pepin KM, et al. (2024) Epidemiological modeling of SARS-CoV-2 in white-tailed deer (*Odocoileus virginianus*) reveals conditions for introduction and widespread transmission. PLoS Comput Biol 20(7): e1012263. <https://doi.org/10.1371/journal.pcbi.1012263>

Minor comments:

1. The repeated use of "human public health" in multiple sections (for example, lines 125 and 148) is redundant. Since "public health" inherently involves human health, simply using "public health" is sufficient and clearer.

We have shortened to only include ‘public health’ as suggested.

2. In line 171, the first use of the abbreviation 'COVID-19' should include its full description.

We have added the full description ‘coronavirus disease 2019’.

Reviewer #2 (Remarks to the Author):

Thank you for the opportunity to review this manuscript. I found the approach and the results interesting, and generally enjoyed reading the manuscript. The article provide an example of the benefits of a coordinated One Health approach to disease management, demonstrated through modeling spillover of SARS-CoV2 from human-to-deer and amplification within deer populations. I think this is a useful contribution. However, there were some issues that will need to be addressed before publication.

My main concerns are related to the adequacy of the model to meet all the challenges presented in the stakeholder engagement section of the manuscript. Clearly, the model is only addressing a fraction of the “fundamental objectives” identified by each sector, which creates a little bit of a disconnect between the two sections. If the model ignores 5 of 8 fundamental objectives, is it really helpful in exploring the decision space and identifying tradeoffs.

We agree that this paper includes an evaluation of only 3 of the 8 fundamental objectives and therefore is limited in its ability to examine and navigate any tradeoffs and identify the best performing alternative; however, including the full decision context of the One Health problem (and not evaluating the full set of objectives) was a conscious decision by the team because we wanted to fully communicate all aspects that might make this decision difficult (i.e., objectives of different agencies) but we also had insufficient detail to estimate effects of the other objectives.

This paper is a critical first step in achieving One Health, which is to document that there are multi-sector actions that agencies can take that are better than the sum of their parts (e.g., the One Health alternative in our paper performed better than the sum of the individual actions that each agency could take alone). We hope that, with the release of this paper, agencies are interested in evaluating the full suite of objectives and grapple with the tradeoffs given the evidence that One Health actions are capable of providing such benefits

to the management of SARS-CoV-2 in deer (and the risks it presents to humans, domestic animals, and other wildlife). To adequately explore tradeoffs and explore the full decision context, a multi-criteria decision analysis (which would involve obtaining performance estimates for the other 5 fundamental objectives under each alternative, as well as decision maker weights) is possible. Our goal with this manuscript was to evaluate the fundamental objectives related to SARS-CoV-2 (#s 1 – 3) to begin exploring the benefits of One Health collaborations on disease dynamics.

The model was, however, very useful in exploring the added value of multi-sectorial management strategies compared to mono-sectoral (the main contribution of this article) in addressing the risk of spillover and amplification. The manuscript should better articulate, the connection between these two sections, and acknowledge and justify that the model is only addressing a narrower scope.

Thank you for this comment and for flagging this important need for clarification. We have added a sentence to acknowledge the narrower scope on lines 262-265.

Further, recognizing the scope of the model, it was not well justified how the measurable attributes were actually aligned with fundamental objectives 1-3. I indeed suspect that they may not always be good proxies for the stated objectives (see detailed comments below). This may all be addressed by using appropriate terms, and stating transparently what the model can and cannot do. i.e with a 120 day simulation, I am not entirely sure that you can assess long-term maintenance in deer populations, however, you can assess initial spillover and amplification (rather than long-term persistence).

We appreciate this comment on alignment between fundamental objectives 1-3. First, we'd like to acknowledge an error in the measurable attribute description in table 1. For objective 1, the measurable attribute should be the *prevalence* of SARS-CoV-2 in deer, not incidence. We have fixed this in the table.

Second, the measurable attributes were generated in collaboration with the guidance committee of agency managers; therefore, these quantities are of direct interest to the decision makers and are of relevance. However, we also acknowledge that the connections were not made explicit in the original submission. We have attempted to improve our justification for each measurable attribute in lines 313-322.

Finally, we do not believe that the modeling framework in this manuscript (or any modeling framework) can accurately estimate the long-term maintenance of SARS-CoV-2 in deer, including distinct variants, over time (see comment by Reviewer #1 about complex RNA dynamics and recombination; our belief is also supported by other human-associated SARS-CoV-2 modeling work that found that projections were most reliable when they were modeled out 4 weeks into the future). This is especially true given the explicit connection within our model between humans and deer, and the requirement that the model also consider dynamics within humans at the same time as deer. As a result, we necessarily limited our evaluation to occur across a 120-day period where humans and deer (and among deer) contacts would be elevated due to hunting activities (and deer mating

behaviors). We maintain use of the word, “persistence”, in our manuscript but define it as applying to the 120-day period rather than over longer timeframes.

The detailed comments below include both minor edits and more substantial comments that align with this general statement:

L48: replace “helps” with “aims to”

Done.

L50: Does “across OH sectors” refer to effort or spread? Clarify sentence

Done.

L50: revise term "risk problem"

Done.

L51: do you need "human" in front of public health?

No. Removed throughout.

L52: Framing of what?

We attempted to clarify this in the preceding sentence by adding “We first used decision analysis methods to frame the One Health problem...”

L58: Here and throughout, I suggest the use of term strategies instead of "alternatives"?

We would like to keep the terminology that is used in the decision analysis literature and therefore would like to use alternatives. We are, however, interested in ensuring that readers understand our meaning and so we have added clarifying language on line 141.

L118: May need to be a little more focused, do you mean health and natural resources, or all agencies?

We have attempted to clarify by adding to the sentence. It now reads –

“Furthermore, U.S. management agencies may be essential to participate in solving One Health problems have governance structures that are complex, with disjunct regulatory authorities that could challenge effective disease management.”

L124: You have used the term sector before, but jurisdictions here... do you mean the same here or ...

We have changed this from ‘jurisdictions’ to ‘sector’ to improve clarity.

L125: What does "This decision framing" refer to?

We have added additional detail to clarify.

The sentence now reads “This decision framing surrounding the management of SARS-CoV-2 transmission between humans and deer and among deer included articulation of sector-specific and shared fundamental objectives, identification of causal chains of interactions that may facilitate spillover and spread, and a specification of possible management alternatives (i.e., management strategies).”

L131: Is "One Health collaborative" referring to the same multi-jurisdiction team? I suggest to call it something and to stick to that name to avoid confusion.

We changed this to multi-sector guidance committee and refer to guidance committee throughout the rest of the manuscript.

L142-145: Move this to discussion

We moved to the last paragraph of discussion.

L145: is this another term for the same group?

Yes, we will use guidance committee to identify this group throughout.

Suppl table 1: It appears that PH mostly had federal folks involved... what is the effect of that on the committee?

Guidance committee member Chelsea Gridley-Smith (Director of Environmental Health at the National Association of County Health Officials) represented the interests of local public health officials surrounding the management of potential zoonotic diseases at scales that are finer than federal. Further, Susan Rollo (State Public Health Veterinarian – Texas) also brought important public health knowledge to the committee even though she primarily provided expertise relevant to state agricultural management agencies.

L149: again, I believe the word "strategies" would be more appropriate here?

We would like to use the term ‘alternative’ to be consistent with decision analysis literature but see reference on line 141 to ‘strategies’.

L165: I suggest to keep the use of the term "framing" consistent in your manuscript. If you describe decision framing as the process of identifying potential strategies, then stick to that and use alternate language to describe other processes. Here you could just replace "framing" with "describing"

We have attempted to use the term framing to describe the specific elements of the decision to be analyzed. That includes the scoping, objectives, alternatives. To make it more clear to the reader, we have added an additional section in the methods to demarcate the framing elements from the modeling work that was used to evaluate the alternatives with respect to the disease-related objectives.

L176-179: This would typically be more of a discussion statement

We have kept this specific description in the methods to provide additional context of the discussions with the committee surrounding the justification for the different objectives.

L169-176 and L179-181: This should be in your result section.

We agree that there are some elements covered in the methods that could be considered results in a different manuscript (e.g., one focused on the guidance committee process and decision framing elements). However, we thought that the modeling methods we present starting on line 310 would lack important context provided by lines 169-176 if we only described the objectives in the results section. Ultimately, we made the decision to present the complete details of the framing work in the methods because we felt that it presented a

better flow and also helped us highlight the main research objectives, which pertain to our modeling results, in the results section.

Table 1: How is "median SARS-cov2 incidence in deer across 120-day period" an indication of infection in humans? I understand that you are trying to match sector-specific objectives with measurable attributes of the mode, but this does not seem like a good match. Being realistic about what the model can and cannot assess is important for this work.

We apologize for the error in this table. The infection in humans should have been measured as the prevalence in deer across 120-days as stated in line 313. The guidance committee selected this measurable attribute because of an assumption that a higher SARS-CoV-2 prevalence in deer would also elevate the potential for any contacts between humans and deer to lead to a higher risk of deer-to-human transmission.

Similarly, not quite sure that a 120-day simulation supports this objective. As noted further down, This model is really focused on spillover and amplification, but is not suitable to assess maintenance within deer populations. Spillover/amplification are extremely important in the process of establishing new reservoirs, and worthwhile assessing with a model, but the authors should be more realistic about what the model can and cannot measure/estimate.

We agree that the model primarily focuses on spillover and amplification patterns under the different alternatives and have therefore removed our discussion surrounding the potential for SARS-CoV-2 to establish as a reservoir that was previously on line 117. We also think that maintenance or persistence (i.e., sustained disease) over 120-days is a useful measure for this particular setting because of the elevated contact rates between humans and deer, and among deer, that may occur during fall/winter months in the U.S. (which was the focus of this study and the period for which we had current and relevant data). During the autumn in the U.S., millions of hunters go afield to pursue harvest opportunities of free-ranging white-tailed deer and also have increased potential for close contact with deer that may lead to intraspecies disease transmission. Further, deer-to-deer contact may also be elevated during this time because of social behaviors that change seasonally (Williams et al. 2014). The evaluation of a longer temporal period would have required the consideration of many additional sources of variation (e.g., contact rates, SARS-CoV-2 dynamics in humans) and uncertainties surrounding the dynamics of SARS-CoV-2 as mentioned previously (e.g., mutation, variant emergence) that we were uncomfortable including.

We would be happy to further re-evaluate/re-word text based on reviewer feedback, and have updated the manuscript to include the 120-day time period whenever we refer to "sustained" spread.

Citation: Williams DM, Dechen Quinn AC, Porter WF (2014) Informing Disease Models with Temporal and Spatial Contact Structure among GPS-Collared Individuals in Wild Populations. PLoS ONE 9(1): e84368. doi:10.1371/journal.pone.0084368

Fundamental objective 4-8 are absolutely not addressed in the model and paper...How is the process and model supporting a One Health approach if 5 out of 8 fundamental objectives are not taken in consideration?

As we mentioned above, we view this work as a critical first step to fully implementing any One Health action because it documents that there are cross-sector actions that agencies can take that not only improve health outcomes within a sector but create benefits that are beyond additive (e.g., the One Health alternative in our paper performed better than the sum of the individual actions).

We also made some revisions to the manuscript based on reviewer 1's comments that might also be helpful to this reviewer comment—

“The model was designed to evaluate the disease-related objectives with the expectation that, if effective One Health alternatives were identified, future analyses could estimate the effect on the remaining objectives given the context where specific actions may be taken.”

We would hope that, with the release of this paper, agencies would be interested in evaluating the full suite of objectives and grapple with the tradeoffs given the evidence that One Health actions are capable of providing such benefits to the management of SARS-CoV-2 in deer (and the risks it presents to humans, domestic animals, and other wildlife).

We also believe that it was important to communicate the full set of objectives that must be considered when deciding among alternatives to counteract any suggestion that this is a simple decision to make. In reality, One Health is challenged by these multiple objectives across agencies such that multi-sector solutions are hard to implement.

L206: replace "of committee members" with "to committee members"

Done. Thank you for flagging this grammatical error.

L221-223: Ignoring demography suggest you are not interested in the long-term persistence of SARS-CoV2 in deer populations, which seems counter to the ideas discussed above related to deer acting as reservoirs for the virus.

Ideally, we would be able to evaluate long-term dynamics over periods longer than 120-days. While this is certainly an interest and would be valuable information to have, we lacked sufficient understanding of the disease process in deer and humans to reliably evaluate the problem at those longer scales—even using expert judgment. As a result, we limited our scope to a period that was useful (because it coincided with meaningful temporal variation in deer-deer and human-deer contact rates) and was long enough to observe disease dynamics as the pathogen was introduced, amplified, and individuals became temporarily immune. Finally, we recognize that this study is not sufficient as evidence of deer as a disease reservoir and have avoided interpreting our findings in that manner. There is only a single mention of the term reservoir in the manuscript, in the introduction, where we bring together lines of evidence that would suggest that deer could be a reservoir for SARS-CoV-2 based on others work.

L227: Is it reasonable to assume a closed population during hunting season? Please justify
Short answer – no. We do not believe it is a closed population, and we recognize that many deer are removed (i.e., harvested) in the season we chose to simulate. However, we also believe that the removals should not affect the conclusions we make in the paper because there is no evidence that harvested deer have higher or lower rates of SARS-CoV-2 infection. Accordingly, we must assume that harvest is random within the population such that the proportion of individuals within the various disease compartments of the SIRS model are harvested with equal likelihood. Based on feedback from the virologists that SARS-CoV-2 deer lack observable symptoms of disease, we felt comfortable making this simplifying assumption.

L244: It was not immediately clear in the objective description that you were going to explore risk uncertainty by simulating across the parameter uncertainty? This may need to be clarified earlier in the objective statement

It's unclear what is meant by the objective statement. We are happy to move around the description of this part of the methods but not sure how to do so.

L247: may be useful to specify that this refers to deer-to-deer R_0 .

Done.

Supplementary Method:

- Missing justification for the time scale of the model... why is it limited to 120 days?

We provide this justification on lines 278-280 in the body of the manuscript.

- Inconsistency in description of mode of transmission: the initial text refer to direct and indirect, but there is no reference to indirect transmission in the rest of the document (instead fluid transmission is also referred to as direct)

We removed reference to indirect transmission.

- reference to equation numbers are not consistent with equation description (equation not numbered)

We removed the numbered references to equations.

- there is no information on how the number of infectious humans is set: I understand it was only considered as a source of infection and not explicitly modeled, but how was it set? A fixed proportion, was it allowed to fluctuate? What was the source of data for it?

We assumed that human COVID-19 prevalence was fixed at 5% based on conversations with the guidance committee and the virology expert panel, however, this parameter was not formally estimated using expert elicited protocols like the other values cited in supplementary table 3. We added this detail to the supplementary table and also it can be found in the Rosenblatt et al. (2024) study cited to support the modeling work.

L258: At this point of the manuscript, it is not clear how the "alternatives" are included in the model

We apologize for not including the details about how the alternatives were included in the model. We have added a table in the supplementary information (Table 3), and a sentence clause that directs the reader there for more detail (line 308). In addition, the vignettes in the public code release provide details on the model and alternatives and allow anyone to reproduce our results. We have added that citation here as well.

L259-263: As indicated above in Table 1: The relevance of measurable attributes to each objectives need better justification. I am not convinced that your measurable attributes are actual indicators of your outcome of interest (i.e. objectives)

We added some detail to this point on lines 313-322 and appreciate the reviewer for pointing out the need for better justification. We expect that this change will also help other readers as well.

L263: Is this evidence of persistence or amplification? What is the evidence that you actually reach an endemic equilibrium? Non-zero prevalence only means that you have continued transmission since the initial spillover and until end of simulation (i.e. persistence ≤ 120 days), it doesn't necessarily mean that the virus will persist endemically in the population.

We acknowledge that we are limited in how we evaluated persistence by the temporal scope of the analysis (120 days) and that we can't say anything about longer term dynamics. For the purposes of this manuscript, we consider equilibrium as the deterministic steady state of the set of ODE equations according to the model structure and the unique set of parameter values with uncertainty used in each of the scenarios. We added this definition of equilibrium to the manuscript.

L264: How did you assess that simulations had reached equilibrium?

For the purposes of this manuscript, we consider equilibrium as the deterministic steady state of the set of ODE equations according to the model structure and the unique set of parameter values with uncertainty used in each of the scenarios. In nearly all simulations, this occurred well in advance of the 120-day window we analyzed. We have added this description on line 323-324.

L266: "summarizes"

Done

L266-267: I think you mean "performance of each alternative relative to the objective considered"

Yes, thank you.

L267: are these tradeoffs? You are not showing any negative effects of any of these interventions, so not sure the term tradeoff is appropriate here.

Yes, this is a good catch. We have removed this language. Thank you.

Fig 2: I am wondering if this would not be better presented as a 3 dimensional graph, where

Pr(H-to_D) and R0 are the main two dimensions, and your third dimension are the different contexts... this would be show the interplay of risk of spillover and risk of subsequent transmission/amplification, similar to the conceptual diagram in Viana et al 2015... Just a suggestion as it may make the graph harder to read

In preliminary analyses, we did explore different ways to plot these figures, but decided on these simplistic representations that convey the information we were interested in.

L297: Clarify your definition here.

Cannot be both a proportion and a number of infections. I think you mean the proportion of the population newly infected during the season. Technically, an incidence proportion (cumulative incidence) should be between 0 and 1. I understand that in this case, you have re-infections, but a more appropriate measurement may be an incidence rate that take into account the appropriate number of individual at risk over time.

We changed this language to incidence rate.

L305: You need to provide a clearer definition of Probability of persistence.

We added the definition to lines 361-362.

Table 2: There is nowhere in the manuscript where you actually explain what the values in the table are. This needs to be better described, how did you derive these values of relative effectiveness? The color shading is also not consistent with the the values, which adds confusion.

The values in the table are explained in the caption for the table and some are reported in paragraphs starting on lines 330 and lines 359. We did not report all of the values in the body of the manuscript and instead refer the reader to the table for the relative performance. We can certainly add more detailed description in text if that's helpful.

L351-353: The promise of OH is about the synergy between sector. In this context, the expectation is that the outcome of joint action is greater than the additive effects of sector-specific actions... I think it would be worth highlighting in your discussion

We have added a sentence clause on lines 424-425 to this point.

L376-380: There should be more specific statements regarding the limitations of your model, particularly in regard to simplifying assumptions (e.g. lack of spatial structure, lack of population heterogeneity etc)

Added on lines 488-490.

L382: There is very little information on how model parameters were elicited in this specific study, which should be better addressed in the material and methods

We would like to cite Rosenblatt et al. (2024) which present specific details about the elicitation process. We have added a citation on line 277.

L385-386: The sensitivity analysis is greatly missing from this article, particularly as a way to assess the effect of parameter choice on your inference regarding different contexts and scenarios.

A full sensitivity analysis, focused on expert elicited parameters, was conducted in Rosenblatt et al. (2024). The reviewers for that paper also identified sensitivity as a missing component and as a result, we conducted a complete sensitivity analysis on the expert-elicited parameters and published those findings as a series of multi-panel figures in the appendix of Rosenblatt et al. (2024) (S14 Fig. and S15 Fig.). We found sensitivity in model results to several parameters related to contact, dose received, and the dose-response relationships and have highlighted those results in the discussion lines 490-493.

To resolve the uncertainties around parameter values is outside of the scope of this manuscript because it would require a better description of the unique setting in which One Health decisions are being made (e.g., for proximity duration and frequency - what is the specific setting for captive and wild deer that would inform estimates of human-deer and deer-deer proximity rates?). Fully describing the context for other objectives would be infeasible for the scale of the current project, which was designed with One Health of SARS-CoV-2 at the National Level (given the breadth and range of agencies that composed the guidance committee).

L394: I am not clear how the manuscript is doing this? Yes, you articulated objectives of each sector, but I am not clear what contradictions were perceived in these objectives.

We are a bit unclear as to what part of L394 the reviewer is referring to with their comment but will try to respond. We believe that we are addressing perceived decision impediments surrounding uncertainty in the efficacy of potential interventions and the ability of individual sectors to decide which actions to take, and governance impediments by articulating the shared decision space of the multiple sectors and evaluating the efficacy of shared actions on disease-associated objectives. We were also careful to use language that suggests that there is more work to do to address decision and governance challenges. (“We begin to address...”).

L399-401: Does this mean this work missed an opportunity to address actual barriers to decision that reach intended outcomes? Are you going to repeat this work? were mechanisms put in place that will facilitate addressing these issues?

As mentioned previously, we viewed this work as a first step in addressing this One Health problem – analyzing whether there are coordinated actions that might enhance what each sector is capable of doing on their own. In order to further address barriers, it is possible to repeat the study with site-specific detail and to quantify the rest of the objectives and navigate tradeoffs using specific decision analysis tools. We hope to be able to do this for SARS-CoV-2 in deer and other One Health problems.

L404-412 should be better referenced with current literature.

We have added an additional citation.

Reviewer #3 (Remarks to the Author):

Overall, I find the article relevant, well constructed, and clearly written. The main result is that adopting a One Health strategy of multi-sectoral expertise and prioritisation, and simultaneous actions to reduce the transmission of SARS-CoV-2 among humans and white-tailed deers, is more efficient than non-coordinated sectoral actions.

The paper provides a quantitative estimate of the impact of the different strategies of action (coordinated or not), based on a model built on the multi-sectoral expertise. It also implements a method to produce multi-sectoral expertise, as well as prioritisation of possible measures. It should emphasise better its methodology as a contribution.

Thank you for the kind comments and seeing the contributions that such a manuscript makes to the literature. We have added more detail on methodology, but are also drafting a separate process-oriented paper that will describe these methods in detail. As a result, we have attempted to cover critical aspects for the readers' understanding in this paper, but were striving for a concise presentation.

The work is relevant to the field of the governance of infectious diseases that display contaminations between humans and domestic / farm / wild animals. The fact that coordinated interventions are more efficient than non-coordinated ones is expected. However, the demonstration contributes to raising the standard for demonstrating the efficacy of the One Health approach. Indeed, the claim that One Health strategies are more efficient than non-coordinated sectoral strategies is frequently made at the international and national levels, but rarely estimated precisely at the sub-national level. The work supports the conclusions and claims.

The text would gain from expressing more clearly, at the start of the paper, how the authors define a One Health strategies, since it first appears in l. 330-331 (“The One Health alternatives (i.e., joint actions taken by all sectors simultaneously) led to the best outcomes” and l. 339-341, “We then considered the effect of coordinated actions by multiple sectors in line with a One Health approach. Thus, we evaluated the outcome when all sectors implemented their own best performing intervention simultaneously”).

We have added language to introduce the idea of One Health alternatives much earlier (lines 147-148). There are now sentences that read --

“Then, we evaluated a range of single-sector and multi-sector alternatives (i.e., “One Health alternatives” that require joint actions by all sectors simultaneously) for their ability to limit the spread of SARS-CoV-2 in white-tailed deer.”

The authors highlight the benefits of a coordinated One Health strategy, but the text does not discuss the costs and difficulties to design and implement it. This is not the core of the paper, and

is difficult to make, but extra thoughts (and data, if applicable) on this would give more breadth to the paper.

This is a great point. We have added a few additional sentences in the discussion (lines 480-484) and have many ideas on this topic. However, we are also sensitive to expanding the manuscript too much. We hope what we have added is sufficient to address this great (and critical) point.

In the same vein, I suggest to change l. 115-116 “Like other zoonoses, managing SARS-CoV-2 transmission between human and non-human hosts is a challenge that requires a One Health approach” → “... that could benefit from a One Health approach”, since the benefit is what is to be demonstrated rather than claimed at the beginning.

We have made this change. Thanks for the suggestion.

The number of workshops, the selection of the participants, and the organisation of the discussions, constitute a sound and coherent methodology.

Epidemiological modelling is outside my area of expertise, therefore I cannot provide relevant feedback on this section. However, it is not the core of this paper.

The concepts used would gain from being made more explicit at the beginning of the text. One main object of the article is governance, for a pluridisciplinary audience, but the writing uses concepts (such as “framing”, “alternatives”) with a meaning that is much more frequent in epidemiology than in political science and could be clarified to ease the reading.

We have attempted to be clear with the meaning of terminology in the manuscript and are a bit hesitant to change away from terms frequently used in decision science literature (so that readers may access this body of literature if they find it useful to work on other One Health problems). However, this is also a comment that we received from another reviewer so there is clearly a challenge with making the work accessible to a broad audience. As a result, we have tried to add in clarification on first use and in captions for the terms mentioned here in hopes of striking a balance.

Examples:

At lines 139 and 141 in the Introduction we added two underlined clauses to the sentences as indicated below.

“This decision framing surrounding the management of SARS-CoV-2 transmission between humans and deer and among deer included articulation of sector-specific and shared fundamental objectives (i.e., basic values that agencies strive to achieve), identification of causal chains of interactions that may facilitate spillover and spread, and a specification of possible management alternatives (i.e., management strategies).”

Table 1 caption

“Fundamental objectives (i.e., principle agency values) ...”

Figure 1 caption

“of alternatives (i.e., strategies) identified across sectors, and the purple hexagons show the collective set of fundamental objectives (i.e., principle agency values) that each of the three One Health sectors may consider when making risk management decisions. The dotted line is the area of the causal chain that we focused on for this risk assessment.”

The methods are generally well presented. The availability and documentation of the code are excellent.

The methodology used to run the meetings with the stakeholders should be explained in greater detail.

How were the discussions structured? Which methodology was used to get to consensual results? For instance, sentences such as l. 165-166 “We then worked on framing the specific interests (i.e. objectives), the causal chains of interactions that may affect disease transmission, and interventions to interrupt transmission” could gain from explaining how the work mentioned was done.

We agree that this would be helpful for the reader, but we are sensitive to the length of the article. Instead of adding more detail to the body of the manuscript, we have provided the meeting schedule that was provided to the guidance committee in the supplementary materials.

Reviewer #3 (Remarks on code availability):

The code is presented as an R package, with excellent documentation. I could install and run the code. I did not test everything, but what I tested worked flawlessly.

Great! We are glad that the code was easy to access and run.

REVIEWERS' COMMENTS

Reviewer #1 (Remarks to the Author):

The authors have adequately addressed all my comments and I have no further feedback on the manuscript

Reviewer #1 (Remarks on code availability):

I checked the code availability, however I did not evaluate the code as it is outside of expertise

Reviewer #2 (Remarks to the Author):

Overall, I am satisfied with the revisions of the manuscript and the justifications that were provided.

A few issues remain.

In my initial review, I pointed to an issue in the terminology related to incidence, as the initial definition was inaccurate. I also suggested that calculating incidence rates that accurately kept track of the population at risk of infection/re-infection would be more appropriate given the ability of individuals to recover and be re-infected, but I certainly did not suggest to just switch the term to incidence rate. The authors are not estimating incidence rates, and the current statement "incidence rate (the average number of infections during the season)" is not accurate. This needs to be adjusted.

I am still not convinced by the justification on L259 that "We assumed that incidence was a good measure for deer health outcomes because it represented the rate of infection for deer and that more infections would result in more negative health outcomes for those individuals." First, the portion of the sentence stating "more negative health outcomes for those individuals" is inadequate as it suggests some notion of individual health outcome, which clearly has more to do with the pathology of infections than the population-level incidence. If population-level impact is really of interest here, I am wondering why the authors would not use the total magnitude of the outbreak (i.e. total number of cases at the end of the 120-day period) rather than an average incidence.

Finally, the content of table 2 is still insufficiently described. Yes, the authors describe in the legend that the values the relative effectiveness, but they provide no information on how these values are derived (or at least I could not find the information).

Thanks

Reviewer #3 (Remarks to the Author):

As with the initial version, I find the article relevant, scientifically sound, and clearly written. Significant improvements have been made to the text, making it more legible and nuanced.

My comments have been addressed, namely:

- The methodology is better explained, I found the extra supplementary material very useful;
- The notion of a "One Health strategy" is defined better and earlier;
- The costs and difficulties of implementing such a strategy are developed to a commendable extent, considering it is not the core of the study, which makes the discussion more nuanced;
- The main concepts of decision science are easier to understand for specialists of neighbouring fields.

Reviewer #3 (Remarks on code availability):

I did review parts of the code in version 1, it worked well as was excellently documented.

I did not review it again for this version, since I did not see the other reviewers mentioning any criticism about it, and therefore assumed the code was unchanged.

REVIEWERS' COMMENTS

Reviewer #1 (Remarks to the Author):

The authors have adequately addressed all my comments and I have no further feedback on the manuscript

Reviewer #1 (Remarks on code availability):

I checked the code availability, however I did not evaluate the code as it is outside of expertise

Thank you. We're glad that the comments of reviewer 1 were sufficiently addressed.

Reviewer #2 (Remarks to the Author):

Overall, I am satisfied with the revisions of the manuscript and the justifications that were provided.

We appreciate the review of our revisions by reviewer 2 and have attempted to clear up the remaining issues.

A few issues remain.

In my initial review, I pointed to an issue in the terminology related to incidence, as the initial definition was inaccurate. I also suggested that calculating incidence rates that accurately kept track of the population at risk of infection/re-infection would be more appropriate given the ability of individuals to recover and be re-infected, but I certainly did not suggest to just switch the term to incidence rate. The authors are not estimating incidence rates, and the current statement "incidence rate (the average number of infections during the season)" is not accurate. This needs to be adjusted.

We apologize for our confusion in what the reviewer was requesting during our last revision. We now understand that they suggested that a different metric may be more useful to measure the efficacy of alternatives to maximize deer health.

In response, we have changed the naming of this metric once more to 'per capita cumulative infections' so that it is more descriptive and meaningful to epidemiologists; it remains, however, the same calculation as 'incidence proportion' described in Rosenblatt et al. (2024).

We have also clarified the definition as follows:

Starting on line 381: "For objective two, we used median per capita cumulative infections of SARS-CoV-2 in wild and captive deer. This metric provided us with the summed total proportion of infections during 120-day simulations. Per capita cumulative infections can

be greater than 1 and represented an outcome where more individuals were infected multiple times than individuals that were not infected within a single simulation ([9]). We assumed that per capita cumulative infections were a good measure for deer health outcomes because this metric quantifies the degree of spread within a population during 120 days, including the potential for immunity to wane and for re-infection to occur.”

We would also like to add a few additional comments of clarification about this measure:

- We chose to keep the measure in its current form so that it is directly comparable to the Rosenblatt et al. paper (2024) and so that it aligns with how these results were communicated to decision makers involved in this project. However, we would like to add that it is possible to track the total number of infections, as the reviewer has suggested, by performing a simple transformation. This can be done by taking the sum across the 120 days of the product of the total population size and the proportion of animals that transition from susceptible to infectious compartments during each time step.**
- The metric does capture infection/re-infection and we clarified our description by adding: “We assumed that per capita cumulative infections were a good measure for deer health outcomes because this metric quantifies the degree of spread within a population during 120 days, including the potential for immunity to wane and for re-infection to occur”.**

I am still not convinced by the justification on L259 that "We assumed that incidence was a good measure for deer health outcomes because it represented the rate of infection for deer and that more infections would result in more negative health outcomes for those individuals." First, the portion of the sentence stating "more negative health outcomes for those individuals" is inadequate as it suggests some notion of individual health outcome, which clearly has more to do with the pathology of infections than the population-level incidence. If population-level impact is really of interest here, I am wondering why the authors would not use the total magnitude of the outbreak (i.e. total number of cases at the end of the 120-day period) rather than an average incidence.

We assumed that this was a good proxy measure for population-level impacts because the more individuals that are infected, the more likely it is that the population experiences some departure from health caused by SARS-CoV-2. The metric, while simplistic, also tracks individuals that are re-infected. Finally, as we mentioned above, it is possible to track the total number of infections, as the reviewer is suggesting, by performing a simple transformation of the current metric. This can be done by taking the sum across the 120 days of the product of the total population size and the proportion of animals that transition from susceptible to infectious compartments during each time step.

Finally, the content of table 2 is still insufficiently described. Yes, the authors describe in the legend that the values the relative effectiveness, but they provide no information on how these

values are derived (or at least I could not find the information).

Thanks

We apologize that these details were still lacking. We have added more description of the measurable attribute for prevalence (line 380-383 in clean draft), and per capita cumulative infections (lines 386-389), and previously improved our description of persistence (line 392-398). We have also referred the reader to Rosenblatt et al. 2024 and Rosenblatt et al. 2023 which includes a full mathematical description of these metrics. Finally, we added a short section of the caption for Table 2 to direct the reader to the methods for better description of the values in the table.

Reviewer #3 (Remarks to the Author):

As with the initial version, I find the article relevant, scientifically sound, and clearly written. Significant improvements have been made to the text, making it more legible and nuanced.

My comments have been addressed, namely:

- The methodology is better explained, I found the extra supplementary material very useful;
- The notion of a "One Health strategy" is defined better and earlier;
- The costs and difficulties of implementing such a strategy are developed to a commendable extent, considering it is not the core of the study, which makes the discussion more nuanced;
- The main concepts of decision science are easier to understand for specialists of neighbouring fields.

Reviewer #3 (Remarks on code availability):

I did review parts of the code in version 1, it worked well as was excellently documented.

I did not review it again for this version, since I did not see the other reviewers mentioning any criticism about it, and therefore assumed the code was unchanged.

We appreciate the review of our revisions by reviewer 3 and are happy that our changes satisfied their initial comments.